# CHEBYUNIT: ENERGY-EFFICIENT FPGA LOW COMPUTATION COMPLEXITY ARTIFICIAL INTELLIGENT ACCELERATOR

## ABSTRACT

Multi-Layer Perceptrons (MLPs) achieve strong accuracy but require large parameter sets, resulting in heavy memory and power consumption. Kolmogorov-Arnold Networks (KANs) mitigate this by replacing weight matrices with learnable functions, yet B-spline implementations remain hardware-complex. To address this, we propose a hardware-efficient Chebyshev-KAN framework that leverages the simple recurrence and numerical stability of Chebyshev polynomials. The proposed ChebyUnit generates polynomial bases and reuses on-chip coefficients to perform lightweight streaming inner-product operations, significantly reducing DDR access and FPGA resource usage. Our Verilog implementation on a Xilinx ZCU102 achieves over 90% reductions in LUT, FF, and DSP utilization compared with an HLS baseline while preserving high approximation accuracy. These results demonstrate that Chebyshev-KANs offer practical efficiency and interpretability, making them well suited for energy-constrained edge AI applications.

## 1 INTRODUCTION

Deep learning continues to advance fields such as image classification, speech recognition, and natural language processing, but the rapid growth of model sizes—often reaching hundreds of billions of parameters [5]—has dramatically increased computational and memory demands, making traditional Multi-Layer Perceptrons (MLPs) increasingly impractical for resource-constrained settings. Kolmogorov-Arnold Networks (KANs) [6; 8] address this issue by replacing large weight matrices with learnable functional bases, reducing parameters while maintaining accuracy and providing clearer interpretability. However, typical KAN implementations using B-spline functions suffer from hardware-unfriendly piecewise structures and complex recursive evaluations. To overcome this limitation, we propose an FPGA-based accelerator for Chebyshev-KAN that leverages the numerical stability and simple recurrence of Chebyshev polynomials [12]. Our design enhances hardware efficiency by adopting weight-reuse mechanisms to reduce memory traffic and minimize LUT and DSP utilization. Implemented in Verilog, the accelerator provides substantial performance and energy-efficiency gains, demonstrating that Chebyshev-KAN is a promising solution for interpretable and low-power edge AI systems.

## 2 RELATED WORKS

### 2.1 KOLMOGOROV ARNOLD NETWORKS

In this work, we adopt the Kolmogorov-Arnold representation[8] , which states that any continuous multivariate function can be written as

$$f(x) = f(x_1, \ldots, x_n) = \sum_{q=1}^{2n+1} \Phi_q \left( \sum_{p=1}^{n} \phi_{q,p}(x_p) \right) \tag{1}$$

From equation (1), $\phi_{q,p}$ and $\Phi_q$ are univariate continuous functions. The theorem provides a constructive scheme for expressing a complicated function as a combination of one-dimensional func-

tions. In this representation, high-dimensional functions can be controlled by analyzing their behavior through single-variable components. For its core, the result states that a function of many variables can be written as a finite sum of terms, each depending on the interpretation of control points . This decomposition is broadly useful in mathematics and applied science because it streamlines both theoretical analysis and practical computation for multivariate models. It shows that continuous multivariate functions are not more sophisticated than univariate functions since they can be made up of one-variable functions. Moreover, we have turned this mathematics theorem into hardware architecture in the following section.

For a layer $l$ with input width $n_l$ and output width $n_{l+1}$, KAN replaces edge weights with univariate edge functions. The $j$-th output coordinate is the sum of those edge activations. In matrix form, the "product" denotes applying each univariate function to the corresponding input and summation along the rows. Then, for layer-$L$ KAN with layer widths $(n_0, \ldots, n_L)$ and a scalar output $n_L = 1$, the network can be written as the following sum of the edge function (3).

$$x_{l+1,j} = \sum_{i=1}^{n_l} \tilde{x}_{l,j,i} = \sum_{i=1}^{n_l} \phi_{l,j,i}(x_{l,i}), \quad j = 1, \cdots, n_{l+1} \tag{3}$$

In matrix form, this represents as follows

$$\mathbf{x}_{l+1} = \underbrace{\begin{pmatrix} \phi_{l,1,1}(\cdot) & \phi_{l,1,2}(\cdot) & \cdots & \phi_{l,1,n_l}(\cdot) \\ \phi_{l,2,1}(\cdot) & \phi_{l,2,2}(\cdot) & \cdots & \phi_{l,2,n_l}(\cdot) \\ \vdots & \vdots & \ddots & \vdots \\ \phi_{l,n_{l+1},1}(\cdot) & \phi_{l,n_{l+1},2}(\cdot) & \cdots & \phi_{l,n_{l+1},n_l}(\cdot) \end{pmatrix}}_{\Phi_l} \mathbf{x}_l \tag{4}$$

$$f(\mathbf{x}) = \sum_{i_{L-1}=1}^{n_{L-1}} \phi_{L-1,i_L,i_{L-1}} \left( \sum_{i_{L-2}=1}^{n_{L-2}} \cdots \left( \sum_{i_2=1}^{n_2} \phi_{2,i_3,i_2} \left( \sum_{i_1=1}^{n_1} \phi_{1,i_2,i_1} \left( \sum_{i_0=1}^{n_0} \phi_{0,i_1,i_0}(x_{i_0}) \right) \right) \right) \right) \tag{5}$$

Kolmogorov-Arnold Networks (KANs) were introduced to improve the parameter efficiency and interpretability of MLPs by replacing fixed activations with learnable univariate functions, typically parameterized using B-splines. Each edge weight becomes a small adaptive function, inspired by the Kolmogorov-Arnold representation, offering greater flexibility than traditional MLPs.

From a hardware perspective, KANs are attractive due to their reduced parameter counts, which implies potential savings in storage and data movement. However, B-spline evaluations are computationally expensive and hardware-unfriendly. Their recursive and nonlinear structure leads to significant LUT, MUX, and decoder overhead, and naive implementations consume substantial resources [14]. Hardware-focused studies further show that B-spline evaluation is complex to map efficiently onto FPGA architectures and often requires careful quantization to remain cost-effective.

Early FPGA comparisons between KANs and MLPs indicate that although KANs are more parameter-efficient, straightforward KAN accelerators may actually use more LUTs and DSPs than MLPs unless basis evaluation and memory layout are carefully optimized. This underscores the need for dedicated hardware-aware KAN design.

## 2.2 CHEBYSHEV POLYNOMIAL

The Chebyshev polynomial basis can be defined by the polynomial form[12] :

$$\begin{aligned} T_0(x) &= 1 \\ T_1(x) &= x \\ T_2(x) &= 2x^2 - 1 \\ T_3(x) &= 4x^3 - 3x \\ T_n(x) &= 2xT_{n-1}(x) - T_{n-2}(x) \quad \text{for } n \geq 2 \end{aligned} \tag{6}$$

The Chebyshev Kolmogorov–Arnold Network (Chebyshev-KAN) is a novel architecture designed to improve the efficiency and accuracy of nonlinear function approximation. By combining the Kolmogorov-Arnold theorem with Chebyshev polynomials, Chebyshev-KAN offers advantages over traditional MLPs. While MLPs require many parameters and often suffer from inactive neurons, Chebyshev-KAN assigns learnable activation functions to edges, allowing the model to achieve lower parameter counts while maintaining or even improving accuracy.

Since edge functions are explicit and one-dimensional, they can be visualized and analyzed directly. This gives Chebyshev-KAN better interpretability than MLPs, especially in scientific and engineering applications to understand learned transformations. In terms of numerical stability and approximation accuracy, Chebyshev polynomials are renowned for their fast convergence and stable approximation. By integrating these polynomials into KANs, the network can achieve higher approximation accuracy, which, in turn, directly optimizes hardware deployment.

Besides, the network can be expressed as

$$y_{bo} = \sum_{i=1}^{\text{input\_dim}} \sum_{j=0}^{\text{degree}} T_{bij} \cdot C_{ioj} \tag{7}$$

We compute the output y by contracting the Chebyshev basis T with the learnable coefficient tensor C using an Einstein summation (einsum), which multiplies each basis value with its corresponding coefficient and sums over the polynomial degree. This replaces weight matrices with lightweight polynomial evaluations. While prior hardware works focus on B-spline KANs and require large LUT/MUX structures, our design leverages Chebyshev recurrence and on-chip coefficient reuse to enable an efficient FPGA implementation. The full hardware architecture is described in the next section.

## 2.3 FPGA Accelerators Design

FPGA-based accelerators exploit inherent spatial parallelism to offload computationally intensive tasks from the host processor. We adopt a hardware-software co-design methodology on a System-on-Chip (SoC) platform, where the CPU manages flexible control and dataflow, while the FPGA fabric instantiates massive parallel processing units to handle heavy arithmetic workloads. In this architecture, the available DSP count varies across platforms and directly dictates the upper bound of arithmetic density.

To maximize system efficiency, we adopt a hardware-software co-design methodology on a heterogeneous System-on-Chip platform. In this architecture, the host CPU (Processing System, PS) is responsible for managing flexible control flows, including data preprocessing, network communication, and runtime scheduling. Meanwhile, the FPGA fabric (Programmable Logic, PL) is configured as a dedicated coprocessor, instantiating massive parallel processing units to handle heavy arithmetic workloads. The communication between these domains is typically bridged by high-bandwidth interconnects, such as AXI protocols, ensuring that the accelerator is kept fed with data.In this architecture, the available Digital Signal Processing (DSP) slices—specifically hardened blocks like the DSP48E2—serve as the critical resource for arithmetic operations (e.g., Multiply-Accumulate). Achieving high throughput requires not only instantiating parallel compute units but also optimizing the utilization efficiency of these scarce resources through strategies like fixed-point quantization and efficient resource sharing.

## 3 Hardware Implementation Approach

In this work, we implement the Chebyshev-KAN accelerator in Verilog hardware description language (HDL) and integrate it using the Xilinx Vivado tool. Verilog enables fine-grained control over datapaths and timing, while Vivado provides simulation, synthesis, and implementation support. All modules are parameterizable—input/output dimensions, hidden size, and fixed-point precision can be configured at the register-transfer level(RTL) level—allowing one hardware template to support multiple network settings. The full design, from the ChebyUnit inner-product engine to the network-

level interconnects, is deployed on a Xilinx ZCU102 MPSoC running at 100 MHz with configurable fixed-point formats.

## 3.1 ACTIVATION UNIT DESIGN

The Activation Unit is the core processing block of the Chebyshev-KAN. It generates Chebyshev basis values, multiplies them with stored coefficients(controlling points), and accumulates the results to produce the output of a single neuron. The module is fully parameterizable—input dimension, polynomial degree, and data bit-width can be adjusted to trade accuracy for resource usage. Internally, each ChebyUnit consists of three components: a Chebyshev basis generator, a control-point buffer, and an inner-product unit, as illustrated in Figure 1.

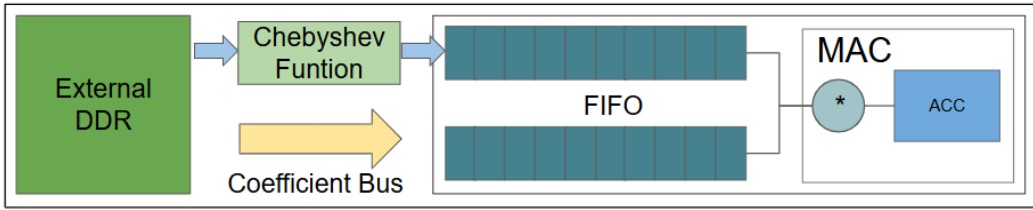

Figure 1: ChebyUnit, single PE structure

### 3.1.1 CHEBYSHEV TENSOR FUNCTION EVALUATION

The first stage of the ChebyUnits is the Chebyshev Tensor Function Generator, which produces the polynomial basis values needed for neuron computation. Given input data $x$, the module computes all Chebyshev terms $T0(x)$, $T1(x)$,...,$Td(x)$ based on the recursive relation mentioned above. In hardware, this recursion is translated into a streaming datapath: The input data x is first registered and its scaled value $(2x)$ is obtained by a one-bit left shift, which is stored for reuse in subsequent computations. This approach eliminates the need to use DSP slices for repeated multiplications by 2, thus saving valuable multiplier resources for other operations. Multiplications in Chebyshev polynomials are mapped to FPGA DSP slices and the simple shift is efficiently handled by basic logic resources.

Once computed, the tensor values are passed one by one, starting from $T0(x)$ up to $Td(x)$. This streaming output allows the next modules, such as the control point buffer and inner-product unit—to begin processing immediately, improving overall throughput without waiting for all terms to finish.

The module operates entirely in fixed-point arithmetic with configurable bit-width and fractional precision, allowing designers to balance accuracy and resource usage. Larger bit-widths improve precision but increase LUT and DSP consumption, while smaller formats reduce area and power. Saturation and overflow are handled by custom arithmetic to maintain numerical stability. With a streamlined datapath that performs iterative recurrence and streams intermediate results, the Chebyshev evaluation module remains compact and efficient, forming the basis of each ChebyUnit.

### 3.1.2 CONTROL POINT STORAGE

The Control Point Line Buffer serves as an intermediate buffer that manages trained parameters during inference. After the model parameters are transferred from external DDR memory to the FPGA through the Direct Memory Access(DMA) interface, they are temporarily stored in this module before computation begins. The storage depth is defined as $degree + 1$, which corresponds to the number of coefficients required to evaluate the Chebyshev polynomial basis functions. During computation, these stored control points are sequentially read out and multiplied with the corresponding polynomial values streamed from the evaluation layer, forming the Multiply Accumulate(MAC) operations that drive the network.

From a hardware implementation perspective, the Control Point Line Buffer is realized using LUT-based distributed RAM. Since control points are repeatedly accessed during inference,

pre-storing them locally avoids repeated DDR fetches, thereby reducing data transfer overhead between memory and FPGA fabric,enhancing the data reuse. This design ensures fast, low-latency access to coefficients and minimizes stalling in the computation pipeline.

## 3.2 CHEBYSHEV KAN NETWORK

Building on the ChebyUnit, the Chebyshev-KAN network instantiates multiple units in parallel based on the chosen hidden size. Although the Kolmogorov–Arnold theorem provides a theoretical upper bound of 2n+1 units for an n-dimensional input, practical models typically require far fewer. The hidden-unit count can therefore be selected according to the complexity of the dataset. As shown in Figure 2, these units operate in parallel, allowing simultaneous polynomial evaluations and inner-product computations, which improves throughput and scalability.

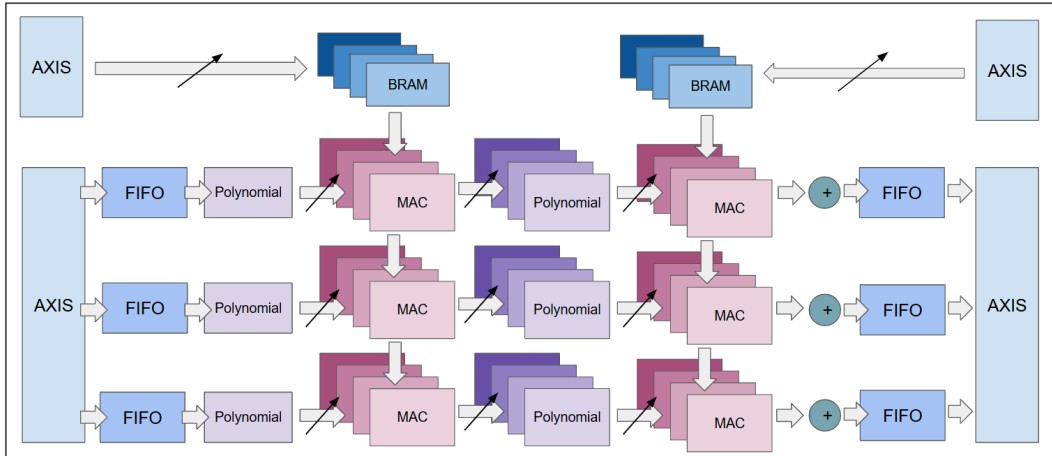

Figure 2: Chebyshev Network Architeture

The control point parameters, trained offline and stored in DDR, are transferred to the FPGA through the DMA engine via the Advanced eXtensible Interface (AXI) Stream (AXIS) interface. These parameters are then distributed by a custom-designed bus structure, which delivers the incoming data to the LUTbased storage inside each ChebyUnit. Using a finite-state machine (FSM) controls this process, determining both the data flow path and whether a given ChebyUnit should accept the incoming data. This mechanism ensures that each ChebyUnit receives the correct set of control points while maintaining efficient data movement across the parallel structure.

Once the control points are loaded, the system enters inference mode. Input data are streamed through AXI-DMA and first stored in a BRAM-based line buffer to overcome DMA bit-width limitations. The buffer reformats the serial stream into parallel inputs, ensuring synchronized delivery to all ChebyUnits. As shown in Figure 3, this process fits into the overall processing-system/programmable-logic (PS/PL) integration architecture, where DDR supplies data, DMA handles transfers, and the custom intellectual-property (IP) performs Chebyshev-KAN computation before returning results to DDR via the symmetric DMA path.

# 4 EXPERIMENTS AND IMPLEMENTATION RESULTS

## 4.1 SETUPS

Using the open-source project provided by SynodicMonth on GitHub[13], we obtain trained parameters of the Chebyshev KAN model, along with its curve fitting results and a performance comparison against a conventional MLP neural network in terms of convergence speed. This setup enables us to compare the numerical results generated in PyTorch with those obtained from the hardware simulation, ensuring that our architecture reproduces the intended computations. The overall workflow is described as follows: In the PyTorch part, the required trained coefficients ($C_{ioj}$, as mentioned in

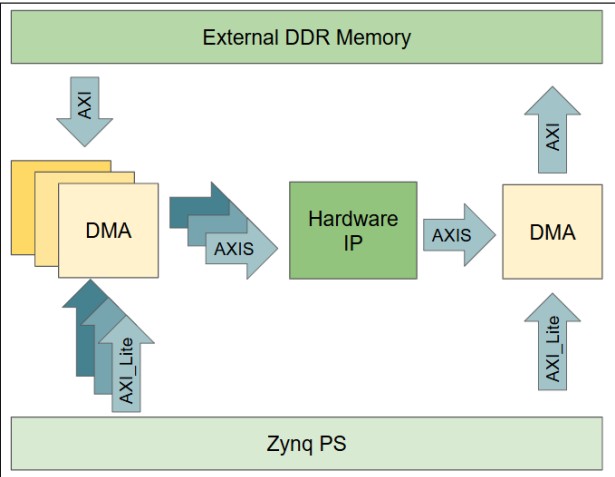

Figure 3: PS and PL configuration

2.2) are extracted from the model. These coefficients are then applied to the hardware part, where the computations are performed. Validation is also conducted using Vivado through simulation, synthesis, and implementation. Finally, the design is deployed on the Xilinx MPSoC UltraScale+ ZCU102 platform, where the actual power consumption and resource utilization are measured[1]. This process enables us to evaluate both the feasibility of hardware implementation and the consistency between software-based and hardware-based computations.

## 4.2 RESULTS

For the following experiment, the input data is defined within the range of [-1, 1]. The test result is visualized using Excel for clarity.

For the target function :

$$f(x, y) = e^{(x^2 + y^2)} \cdot \sin\big(2\pi(x^2 + y^2)\big)$$

It is carried out with an input dimensionality of 2, a polynomial degree of 8, 4 hidden units, and coefficients represented in the Q16.16 format (32 bits). Note that due to the limited number of test data points, the plotted results may appear less smooth when visualized in Excel. Nevertheless, even with the current sampling density, numerical verification confirms that the computed values are correct and consistent with the expected results.

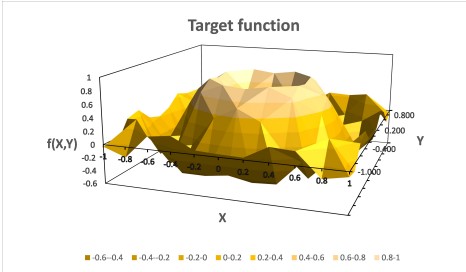 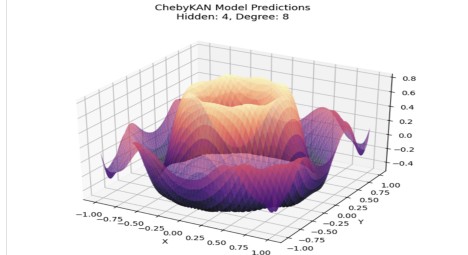

Figure 4: Target Function Implementation      Figure 5: Training Result

The experimental results confirm that the hardware implementation accurately reproduces the expected numerical outputs, demonstrating the correctness and reliability of the design. The primary factor affecting numerical precision is the chosen fixed-point format. Bit-width determines both resolution and dynamic range, creating a trade-off between hardware cost and accuracy: wider formats improve approximation quality but require more logic, registers, and power, while narrower formats

reduce resources at the cost of precision. This highlights the need for careful selection of fixed-point precision when deploying learning models on hardware.

## 4.3 RESOURCE AND POWER ANALYSIS

To better highlight the efficiency of our design, we report both the utilization of hardware resources and the percentage reduction compared to the HLS baseline[14]. It is worth mentioning that the architecture of the HLS implementation differs from ours, as it is based on B-spline to construct each learnable activation functions of the KAN[8]. Table 1 presents the actual resource utilization for each case, while Table 2 reports the reduction percentage, calculated as : [1] : The utilization of resources in HLS implementation[14].

In Table 2, our method achieves significant reductions in LUTs, FFs, and DSPs, with savings exceeding 90% in datasets such as Dry Bean and Mushroom. These datasets belong to standard classification tasks. The Chebyshev-based design (abbreviated as Chebyshev in Table 1), due to its simpler function expression compared to the B-spline based design (abbreviated as B-spline in Table 1), requires fewer hardware resources, especially DSPs, and thus holds a significant advantage in hardware computation. Furthermore, as shown in Table 1, our design consistently achieves a latency of 13 cycles and demonstrates better stability in power consumption. Even for larger model sizes, our approach exhibits lower power usage. To sum up, these results indicate that our method delivers better energy efficiency and data transfer performance.

| Dataset | Model size | Type | Freq (MHz) | Hardware Resources | | | Power (W) | Latency | |
|---|---|---|---|---|---|---|---|---|---|
| | | | | LUTs | FFs | DSPs | | Cycle | Time (ns) |
| Moons | 2,2,1 | B-spline | | 17877 | 8622 | 120 | 0.717 | 128 | 1280 |
| | | Chebyshev | | **9888** | 12150 | **40** | 3.034 | **13** | **130** |
| Wine | 13,4,3 | B-spline | | 146843 | 74741 | 960 | 1.349 | 688 | 6880 |
| | | Chebyshev | 100 | **30154** | **22104** | **324** | 3.293 | **13** | **130** |
| Dry Bean | 16,2,7 | B-spline | | 1677558 | 734544 | 9111 | 14.802 | 1896 | 18960 |
| | | Chebyshev | | **27359** | **25198** | **256** | 3.271 | **13** | **130** |
| Mushroom | 8,24,2 | B-spline | | 3112275 | 1337291 | 16299 | - | 3434 | 34340 |
| | | Chebyshev | | **80393** | **38985** | **1088** | 3.809 | **13** | **130** |

Table 1: Actual Resource Utilization Under Different Datasets and Implementations types

- : Power consumption is not included, as the hardware requirements surpass the FPGA board capacity described in [14].

| Dataset | LUTs Reduction | FFs Reduction | DSPs Reduction | Latency Reduction |
|---|---|---|---|---|
| Moons | 44.69% | - | 66.67% | 89.84% |
| Wine | 79.47% | 70.43% | 66.25% | 98.11% |
| Dry Bean | 98.37% | 96.57% | 97.19% | 99.31% |
| Mushroom | 97.42% | 97.08% | 93.32% | 99.62% |

Table 2: Utilization and Latency Reduction Result (%)

- : In a small-scale model, the synthesizer often implements temporary buffers directly with flip-flops instead of other larger memory blocks.

| Dataset | Bspline | Chebyshev |
|---|---|---|
| Moons | 96.8% | 99.8% |
| Wine | 97.3% | 94.8% |
| Dry Bean | 91.9% | 92.4% |
| Mushroom | 55.8% | 99.6% |

Table 3: Testing Accuracy Result (%)

| Dataset | Bspline | Chebyshev |
|---------|---------|-----------|
| Moons | 36 | 30 |
| Wine | 448 | 320 |
| Dry Bean | 414 | 460 |
| Mushroom | 2160 | 1200 |

Table 4: Number of Coefficients for Different Types

Based on the results presented in Table 3 and Table 4, Chebyshev demonstrates more stable and higher accuracy, confirming the correctness and reliability of our hardware inference architecture. Moreover, Chebyshev requires significantly fewer coefficients compared with B-spline, implying reduced computational demand. Overall, given the low computational complexity and high through-put of the proposed hardware architecture, we believe it has strong potential to deliver outstanding performance even when scaled to more complex and computation-intensive applications.

For the (2,2,1) model, Figure 6 shows the power distribution across system components [9]. The Processing System (PS Dynamic) accounts for most of the power consumption, while the pro-grammable logic—clocks, signals, BRAM, and DSPs—contributes only a small portion (grouped as "Others"). This indicates that internal Programmable Logic (PL) activity consumes relatively little power, yet still performs the core computation efficiently, demonstrating the energy-efficient nature of the FPGA implementation.

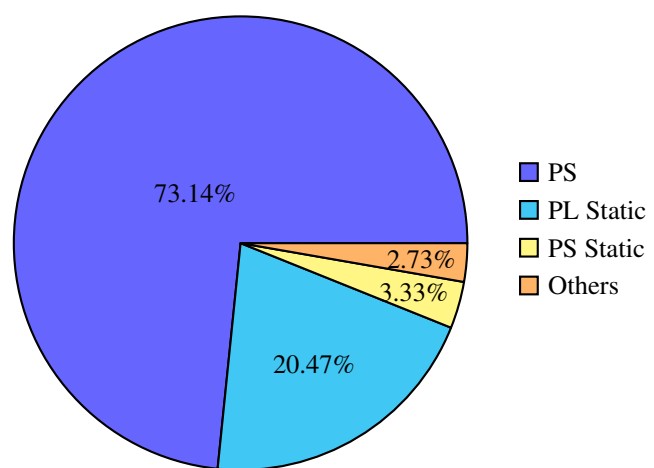

Figure 6: Power Consumption Distribution of the (2,2,1) Model Size

## 5 DISCUSSION AND CONCLUSION

In this work, we develop an FPGA accelerator for Chebyshev-Kolmogorov–Arnold Net-works (Chebyshev-KAN), targeting the challenge of deploying parameter-efficient yet hardware-unfriendly models on energy-limited platforms. By leveraging the simple recurrence and numerical stability of Chebyshev polynomials, the proposed modular ChebyUnit achieves substantial reduc-tions in LUT, FF, and DSP usage while maintaining high approximation accuracy [10; 7]. Experi-ments on the Xilinx ZCU102 demonstrate correct functionality and over 90

A key insight is the precision–efficiency trade-off: wider fixed-point formats improve accuracy but increase DSP and BRAM cost, while narrower formats reduce hardware usage at the expense of fidelity. These observations motivate future exploration of adaptive or mixed-precision schemes [4; 2] to further optimize the balance between accuracy and efficiency.

The design also shows strong scalability. Although deeper or wider networks introduce memory bandwidth and on-chip storage pressure, the integration of weight reuse [11] effectively reduces

redundant transfers and resource demand, providing a clear path for scaling Chebyshev-KAN accelerators.

Thanks to its lightweight functional representation, the accelerator is well suited for edge AI scenarios, reducing parameter storage, computation, and memory traffic while minimizing DSP consumption and off-chip access. This makes Chebyshev-KAN promising for IoT, biomedical workloads [3], and real-time sensing systems.

Limitations remain: evaluations are restricted to small-scale function approximation tasks and a single FPGA device. Future directions include applying the accelerator to larger vision, speech, and time-series workloads, performing systematic quantization analysis, and validating the design across diverse FPGA families or ASIC prototypes.

Overall, our results show that Chebyshev-KAN is both theoretically elegant and practically deployable. FPGAs offer an effective path toward efficient, interpretable neural networks tailored to low-power edge environments.

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
