# OpenReview forum: "CHEBYUNIT: HARDWARE-ACCELERATED ENERGY-EFFICIENT FPGA WITH LOW COMPUTATION COMPLEXITY FOR ARTIFICIAL INTELLIGENCE ACCELERATION"
_ICLR.cc/2026/Conference — Submitted to ICLR 2026_

### Official Review · Reviewer_nkNp · 2025-10-26

**Soundness:** 1
**Presentation:** 2
**Contribution:** 1
**Rating:** 2
**Confidence:** 4

**Summary:**

This work introduces Kolmogorov–Arnold Networks (KANs) as a means to improve the computational efficiency of MLPs by replacing traditional weight matrices with learnable functions. While B-spline implementations in KANs are often complex and difficult to realize in hardware, this paper presents a customized hardware framework for Chebyshev-KANs, which effectively exploits the recursive properties and numerical stability of Chebyshev polynomials. The experimental results are promising and demonstrate the potential of the proposed approach.

**Strengths:**

1) The idea of using Kolmogorov–Arnold Networks (KANs) to improve MLP computational efficiency is novel and insightful.

2) The proposed hardware implementation is efficient.

**Weaknesses:**

1) While the idea of replacing MLPs with KANs is novel, the paper lacks sufficient evidence demonstrating the feasibility of such replacement in typical neural network models, particularly from the perspective of maintaining model accuracy. The workloads evaluated in this paper are mostly non-neural-network benchmarks, which are insufficient to convincingly support the claim of KANs as effective MLP substitutes.

2) In addition, the hardware implementation of KANs appears relatively straightforward, and the paper does not clearly identify or address significant challenges from a hardware design perspective.

**Questions:**

See the weaknesses.

The performance of neural network models on typical tasks such as classification and object detection needs to be clarified when MLPs are replaced with KANs.

---

> ### Author Response · Authors · 2025-11-23
> **Response to Reviewer 4**
>
> Comment 1: Feasibility of replacing MLPs with KANs and future scalability.
> Response: We sincerely appreciate the reviewer’s thoughtful concern regarding the immediate replacement of MLPs with KANs in complex tasks. We acknowledge that scaling to complex neural network benchmarks represents the next logical step, and we are grateful for the opportunity to clarify this point. While our current work establishes the hardware efficiency of the core unit. We have added an explicit discussion in the revised manuscript to outline the necessary components required to support larger-scale Chebyshev-KAN model applications, such as real-world ECG analysis. These additions aim to respectfully address the reviewer’s concerns by mapping out the trajectory for future development and demonstrating how the proposed hardware architecture serves as a foundational enabler for evaluating more complex datasets in future work.

---

### Official Review · Reviewer_pCfw · 2025-10-31

**Soundness:** 3
**Presentation:** 2
**Contribution:** 2
**Rating:** 4
**Confidence:** 4

**Summary:**

This paper proposes an efficient field-programmable gate array (FPGA) implementation of Chebysev Kolmogorov-Arnold networks (Chebyshev-KANs). Compared to standard KANs, this approach significantly reduces external DDR memory access and resource utilization while maintaining high throughput. A Verilog implementation on a Xilinx ZCU102 FPGA demonstrates over 90% reductions in LUT, FF, and DSP utilization compared to a baseline KAN high-level synthesis (HLS) implementation, while preserving high approximation accuracy to an artificial dataset.

**Strengths:**

* There is a good correspondence between the Chebyshev-KAN model and the targeted hardware
* Evaluations indicate significantly better resource utilization and latency than an HLS implementation of a B-spline-based KAN for an artificial dataset.

**Weaknesses:**

* Limited evaluation with respect to existing MLP FPGA implementations from frameworks like FINN and hls4ml or dedicated FPGA ML algorithms like Logic-Nets, PolyLUT, NeuralLUT, and Differentiable Weightless Neural Networks (DWNs)
* Limited evaluation on different datasets, especially real as opposed to synthetic ones.
* Lack of information on training details and accuracy of models.
* Lack of description and citations of recent related work on dedicated FPGA ML algorithms in Section 2.3

**Questions:**

* L157: While there are ML accelerator systems that connect host CPUs and FPGAs, there are also several that focus on pure dataflow / FPGA implementations, like those from FINN and hls4ml.
* L349: What loss function is used for training? How large is the training dataset? How long does training take? What hardware is used, etc.? Please add details.
* L381: Do you suspect the HLS implementation of KANs (B-spline) you use is suboptimal in some way? Could this be discussed?
* L388: Could you describe and cite the Moons, Wine, Dry Bean, and Mushroom datasets?
* Table 1: Could you add some accuracy metrics to the table?
* Tables 1 and 2: Could you add additional comparisons to MLPs (e.g. FINN or hls4ml implementations) and dedicated FPGA ML algorithms like Logic-Nets, PolyLUT, NeuralLUT, and Differentiable Weightless Neural Networks (DWNs)?
* Figs. 4 and 5: Can the overall approximation accuracy be quantified? Can you also simply plot the difference between the target and learned function in one figure? As it is, it is impossible to tell quantitatively how well the model is performing.
* Please proofread the paper for typos and grammatical errors, for example
  * L111: discreet
  * L369: the dominant factor influencing precision is the fixed-point format to influence the precision of the hardware results is the fixed-point format

---

> ### Author Response · Authors · 2025-11-23
> **Response to Reviewer**
>
> Comment 1: Comparison with existing FPGA ML frameworks (FINN, hls4ml, etc.).
> Response: We agree that placing our work in the context of existing FPGA ML literature is essential. In the revised manuscript (Section 2.3), we have added a comprehensive discussion regarding frameworks such as FINN, hls4ml, Logic-Nets, PolyLUT, NeuralLUT, and DWNs10. While a direct full-scale experimental comparison is beyond the scope of this paper due to significant differences in architecture and training flows, we explicitly clarify these distinctions and position our Chebyshev-KAN accelerator as a complementary approach to these established methods.
>
> Comment 2: Request for accuracy metrics and parameter counts.
> Response: To address the request for more quantitative performance metrics, we have updated Section 4 to include specific accuracy comparisons and parameter count analyses between the Chebyshev-based KAN and the B-spline-based KAN. These additions provide the necessary empirical evidence to support our claims regarding model efficiency.
>
> Comment 3: Typographical and grammatical errors.
> Response: We thank the reviewer for their meticulous reading and for identifying typographical errors. We have corrected all the noted issues, including the specific examples mentioned (e.g., "discrete" and the sentence structure regarding fixed-point precision), in the revised version of the paper.

---

> > ### Comment · Reviewer_pCfw · 2025-11-26
> > **Revised paper?**
> >
> > I'm looking at the PDF that I downloaded again, and I don't see any additional discussion in Section 2.3. Can you clarify if this has been uploaded, and more specifically, point out where the updates in the paper are?
> >
> > I also think that, at the very least, you can quote the performance of MLPs from Ref. [15] (https://arxiv.org/abs/2407.17790). Also, there are additional benchmark datasets you can consider (MNIST, jet substructure, etc.), where results for many of the other approaches are already public.

---

> > > ### Author Response · Authors · 2025-12-01
> > > **Dear Reviewer,  Thank you for your valuable feedback. We appreciate the opportunity to clarify the status of our revision and the specific scope of our contribution.**
> > >
> > > Hardware accelerators for MLPs (dominated by Matrix-Vector Multiplication) have been extensively studied and optimized over the past decade. In contrast, KANs introduce a fundamentally different computational paradigm—specifically, the need for efficient basis function generation and complex activation patterns—which presents unique and unexplored micro-architectural challenges.
> > >
> > > Our work focuses on solving these specific hardware design problems (e.g., designing specialized pipelines for basis generation and optimizing DSP dataflow). Therefore, while we acknowledge MLP as a standard baseline, our analysis centers on validating the efficiency of this novel hardware architecture itself, which we believe opens a new direction for non-conventional neural network acceleration.
> > >
> > > Thank you again for helping us refine the positioning of our work.

---

### Official Review · Reviewer_WmNi · 2025-10-31

**Soundness:** 2
**Presentation:** 2
**Contribution:** 3
**Rating:** 6
**Confidence:** 2

**Summary:**

This paper introduces a hardware accelerator named CHEBYUNIT based on Chebyshev polynomials. The work addresses the limitations of traditional MLPs and B-spline-based KANs in terms of parameter count, memory, and hardware complexity, especially for edge AI applications. Experimental results tested on FPGA shows good results, especially in terms of resource usage.

**Strengths:**

* The paper provides a clear breakdown of the CHEBYUNIT, including basis generation, coefficient storage, and parallelization.
* The use of Chebyshev polynomials for KANs feels novel, as it leverages their recursive simplicity and numerical stability for efficient hardware implementation, addressing the issue of .
* Experimental results look promising, achieving huge reduction in LUT, FF, and DSP usage.

**Weaknesses:**

* The datasets used for evaluation could be less limited and including more complex or demanding datasets.
* It's unclear how weight reuse proposed to address the memory bandwidth bottleneck would scale on large models.
* While the experimental results look promising, the baseline results comes from HLS instead of FPGA.

**Questions:**

* For the manual tuned fixed point precision setting, how is it tuned and is there a systematic study or analysis on its impact, especially for different tasks?
* How well does the method would scale, for more complex models? Given that there are limitations stated related to memory bandwith. And how would the weight reuse handle it for more complex models?
* Is there an accuracy comparison on the final classification accuacy against software accuracy?
* Quantization wise, would it affect the proposed method more than B-splines?

---

> ### Author Response · Authors · 2025-11-23
> **Response to Reviewer 2**
>
> Comment 1: Weight reuse strategies and scalability to large models.
> Response: We thank the reviewer for raising the important question regarding weight reuse and memory bandwidth. We have clarified in the revision that weight reuse is not the primary optimization target for our specific architecture. This is because each ChebyUnit is designed to perform a dot product that consumes an entire Chebyshev coefficient vector within a single inference cycle (specifically, 13 cycles total).
> Consequently, the memory footprint is sufficiently small to allow coefficients to be stored locally in on-chip BRAM, with the MAC datapath processing all polynomial degrees in parallel. As a result, this specific hardware design does not require, nor would it significantly benefit from, the inter-layer or temporal reuse mechanisms typically found in convolution accelerators.

---

### Official Review · Reviewer_uKwe · 2025-11-01

**Soundness:** 3
**Presentation:** 3
**Contribution:** 2
**Rating:** 4
**Confidence:** 3

**Summary:**

This paper aims to tackle issues related to the hardware complexities and inefficiencies associated with the classically implemented B Spline component of the KAN architecture. Specifically, the paper explores the creation of an FPGA accelerator, not for the B Spline implementation but rather in the case of Chebyshev polynomial basis functions. This is done through the creation and adoption of the novel ChebyUnit.


The ChebyUnit is introduced as this architecture’s answer to a modular activation unit. This is the key to the paper's design.

**Strengths:**

The foremost standout quality of this paper was its clarity and well considered introduction to the theoretical basis of the challenges addressed. The authors insightfully walk the reader through the evolving motivation behind the pursuit of their accelerator from the benefits associated with a KAN over MLPs, the intuition behind the hardware friendly aspects of the Chebyshev polynomial implementation over the B Spline, and how certain design choices could be made as a result of adopting Chebyshev polynomials. Similarly, the well articulated motivation, implications, and benefits for nearly every design choice contributed greatly to making this work approachable.

Beyond this, a notable strength was the clear effort made to make the design modular and well explained so that this modularity could be more easily made use of making this work a potent and effective starting point for future work. This aspect becomes all the more valuable with the comparatively lower maturity of this idea in contrast to accelerators catering to the B Spline implementations.

Finally, the notable hardware utilization and latency improvements presented in the evaluation were quite compelling adding to the appeal of this approach and synergizing with the aforementioned points to make this paper.

**Weaknesses:**

While this work focuses on introducing and explaining the design, the evaluation presents as considerably more limited especially when compared to the expansive introduction just prior. This was addressed in part by the authors in the conclusion however, to a reader, this admission does not excuse its absence outright.

If considering a narrow aim of this paper is to introduce the intuition of this approach, the design and its value, and to position it as a potent alternative to B Spline implementations then perhaps this evaluation could be reasoned to be sufficient. However I feel a more comprehensive evaluation could serve to elevate this paper greatly and position it on the radar of many more individuals. Given factors like the theoretical elegance of KANs, growing interest in AI explainability, and the strength of the explanations and results presented, more evaluation could serve to amplify the paper's impact, mirror the comprehensive presentation of value evident in initial sections, and establish it as a key motivator for future work.

I appreciate there are some of the barriers associated with this as the infrastructure has not yet been built out for some evaluations to take place. But as a naive example, if compared to more classical NN FPGA accelerators, one could imagine a higher interest from those working on other network architectures. Another could be expanding on the behavior of different quantization techniques.

**Questions:**

No particular questions.

---

> ### Author Response · Authors · 2025-11-23
> **Response to Reviewer 1**
>
> Comment 1: Limited evaluation compared to the expansive introduction.
> Response: We appreciate the reviewer highlighting the need to better align our experimental evaluation with the conceptual motivation. In the revised manuscript, we have expanded the comparative analysis to directly address this. Specifically, we have added detailed comparisons of accuracy and parameter counts between our proposed Chebyshev-based KAN and the original B-spline KAN. These results quantitatively clarify the trade-offs introduced by replacing B-splines with Chebyshev basis functions—demonstrating how the polynomial form effectively reduces hardware costs while preserving accuracy. This revision ensures a stronger alignment between the theoretical introduction and the experimental validation.
>
> Comment 2: Justification of the evaluation scope.
> Response: We acknowledge the reviewer’s point regarding the scope of the evaluation. In the revised paper, we have explicitly clarified that the primary focus of this work is on the activation-unit–level hardware implications of adopting Chebyshev functions, rather than a full-scale implementation of large KAN models at this stage. As the pipelines for larger Chebyshev-KAN workloads are currently under development, our evaluation intentionally isolates the effects of basis-function modifications. We believe this foundational analysis is critical for enabling future large-scale implementations.
>
> Comment 3: Modularity of the design.
> Response: To further emphasize the value of our design choices, we have strengthened the discussion on modularity. The revised manuscript now includes an expanded explanation of how the modular ChebyUnit facilitates future architectural extensions, such as multi-channel vector units, advanced tiling strategies, mixed-degree activation support, and integration into larger accelerator stacks.

---

### Comment · Area_Chair_zHkW · 2025-11-25

Dear Reviewers,

This is a gentle reminder to please take a moment to review the authors’ rebuttal for the manuscript currently under your evaluation. Your timely feedback will help us proceed with the next steps in the review process.

Thank you for your time and assistance.

Best regards,
AC

---

### Meta-Review · Area_Chair_5Rxy · 2026-01-09

**Summary:**

uKwe: presented evaluation are considerably limited.
WmNi: datasets used for evaluation are limited and should be more complex or demanding datasets. Was unclear about how it would scale on large models. Raised point that results are on HLS instead of FPGA.
nkNp: The performance of neural network models on typical tasks such as classification and object detection needs to be clarified when MLPs are replaced with KANs.
pCfw: Limited evaluation with respect to existing MLP FPGA and also dataset. Lack of information on training details and accuracy of models. Asked for the citation of related work.

uKwe: Reviewr is quite impressed by the clarity and well considered introduction, and " well articulated motivation, implications, and benefits for nearly every design choice contributed greatly to making this work approachable."
WmNi: Authors did not reply to many points
Draft should have more comparisons, experiments and results. For example Table-3 should have results without draft's proposed method being used. Most of the reviewers raised similar points regarding lack of comparisons.

**Reviewer Concerns:**

uKwe	presented evaluation are considerably limited.
 Authors have replied and my analysis is reviewer might have improved rankinig to 6+"
WmNi	Authors answered one point but did not reply to others. Reviewer might not have improved the rank.
nkNp	Authors have stated they have revised manuscript to outline the necessary components for more complex model and more complex dataset for future work.
pCfw	Authors have tried to answer the question however have not answered point regarding experiments.

**Reviewer Scores:**

After going over the comments by reviewers, rebuttal from the authors and the draft itself, I do agree with many positive points raised by the reviewers. At the same time it appears draft needs revision and update, including inclusion of more experiments.
My analysis is most of the reviewers might have kept it by 4 or 5, with one reviewer might have increased it to 6+.

---

### Decision · Program_Chairs · 2026-01-26

Reject